# Preparation of Mannitol-Modified Loofah and Its High-Efficient Adsorption of Cu(II) Ions in Aqueous Solution

**DOI:** 10.3390/polym14224883

**Published:** 2022-11-12

**Authors:** Guangtian Liu, Jianjian Liang, Jie Zhang

**Affiliations:** Hebei Key Laboratory of Heavy Metal Deep-Remediation in Water and Resource Reuse, Hebei Key Laboratory of Applied Chemistry, School of Environmental and Chemistry Engineering, Yanshan University, Qinhuangdao 066004, China

**Keywords:** mannitol, loofah, Cu(II) ions, adsorption isotherms, kinetics, regeneration

## Abstract

Adsorption is considered the most favorable method for heavy metal removal. In this paper, a low-cost, high-efficiency heavy metal adsorbent, mannitol-modified loofah (MML) was prepared. Some characterization methods are used to characterize the structure of MML, such as Fourier transform infrared spectroscopy (FTIR), X-ray diffraction (XRD), and scanning electron microscopy (SEM). The adsorption behavior of MML for Cu(II) ions was explored under different conditions, such as the amount of adsorbent, pH, initial concentration of Cu(II) ions, and adsorption time. The results indicated that the adsorption capacity of MML for Cu(II) ions was greatly improved. When the initial concentration of Cu(II) ions was 900 mg/L and the pH is 5.0, the adsorption capacity (Q_e_) was 888.9 mg/g at 298K, which was significantly higher than that of some other modified cellulose adsorbents. Isothermal adsorption results showed that the adsorption process was consistent with the Freundlich model. The adsorption kinetics conformed to the pseudo-second-order equation. Furthermore, the regeneration capability of MML indicates that MML is a cheap and excellent adsorbent for Cu(II) ions removal in wastewater treatment.

## 1. Introduction

In recent years, the heavy metal pollution of water also shows a deepening trend because of the rapid development of the modern industry. Heavy metal wastewater has been paid more and more attention to because of its serious harm to human health. Copper is widely used in metallurgy, electrical, and other industrial production, and it is also an essential element in the human body. However, an excessive intake of copper can lead to brain, liver, respiratory and nervous system diseases, especially in children [1]. Therefore, it is very necessary to remove residual copper ions in water. Many heavy metal treatment methods are applied [2,3,4,5,6], such as precipitation, ion exchange, electrodialysis, adsorption, etc., among which the adsorption method is an environmentally friendly and low-cost method for the removal of heavy metal ions from contaminated water.

Agricultural and forestry waste is rich in raw material sources, has low-cost, and has good adsorption effects. It has a broad application prospect in the field of heavy metal pollution remediation. In recent years, there have been many related studies [7,8,9,10,11,12,13,14,15]. Lee et al. [7] reported the adsorption of heavy metal ions from an aqueous solution by persimmon biscuit. The results showed that the maximum adsorption capacity of persimmon biscuit to Cu(II), Pb(II), and Cd(II) was19.42 mg/g, 22.59 mg/g, and 18.26 mg/g, respectively. Park et al. [8] investigated the adsorption properties of the pepper stem biochar. The results showed that the maximum adsorption capacity of Pb(II), Cr(II), Cd(II), Cu(II), and Zn(II) on the pepper stem biochar was 131 mg/g, 76 mg/g, 67 mg/g, 48 mg/g, and 31 mg/g, respectively. Nada et al. [9] employed banana leaves as an adsorbent to remove metal ions from wastewater. However, the disadvantage of agricultural and forestry wastes istheirlow adsorption capacity for heavy metals. Therefore, in order to improve their adsorption capacity, various modifiers can be used, such as ethylenediamine [16], acrylic acid [17], and ethylenediamine tetraacetic anhydride [18].

Loofah is a kind of agricultural waste, consisting of cellulose, hemicellulose, and lignin, and is the fibrous vascular reticulated system of the matured dried fruit of Luffa cylindrical [19]. The monolith-like supermacroporous structure of loofah makes it an ideal adsorbent for heavy metal. It contains highly active groups, which can be chemically modified to further improve the adsorption active site. Mannitol is inexpensive, readily available, water-soluble, and contains multiple hydroxyl groups. In this paper, a large number of epoxy groups were introduced onto the surface of the loofah by grafting a PGMA chain, which greatly increased the probability of functionalization withmannitol. Compared with other modification methods [20,21,22], it is more advantageous to increase the amount of hydroxyl, and then improve the adsorption capacity of loofah to heavy metal. It is expected to be widely used in the field of heavy metal wastewater treatment.

## 2. Experimental

### 2.1. Materials

Loofah was collected from Qinhuangdao, China. Glycidyl methacrylate (GMA) was purchased from Shanghai Baishun Biotechnology Co., Ltd., Shanghai, China. HCl, CuSO_4_·5H_2_O, the emulsifier(OP-10), K_2_S_2_O_8_, 1,4-dioxane, sodium bisulfate, and mannitol were provided by Tianjin Dengfeng Chemical Reagent Factory, Tianjin, China.

### 2.2. Synthesis of Mannitol-Modified Loofah (MML)

The cleaned loofah was cut into small pieces with scissors and then pulverized with a high-speed pulverizer until powders smaller than 74 μm were obtained. Experimental steps were as follows: 4.0 g of loofah was put into a solution of water containing an emulsifier (0.3 g) for 30 min at 40 °C. Then, 8.0 g GMA, 1.5 g K_2_S_2_O_8_, and 2.0 g NaHSO_3_ were added and the mixture was stirred using a constant temperature magnetic heater stirrer for 80 min at 35 °C. Finally, the product was washed with acetone and ethanol and dried at 80 °C; loofah-g-GMA was obtained.

An amount of 1.0 g of loofah-g-GMA was added to 40 mL1,4-dioxane solution to swell for 6 h in order to fully dissolve the PGMA chain in 1,4-dioxane, then 10 mL of deionized water was added (V_Water_:V_1,4-dioxane_ = 1:4), 12.0 g mannitol was added slowly at the same time and stirred at 85 °C for 20 h. The product was filtered and extracted until neutral with deionized water, and the solid was then dried in a vacuum. MML was obtained. The principle of modification is shown in Figure 1.

### 2.3. Characterization Methods

Functional group information was characterized by Fourier transform infrared spectrometer (FTIR) (Nicolet380 Spectrometer System, Nicolet Company, Mountain, WI, USA). The crystal structure of loofah, loofah-g-GMA, and MML was characterized using a Rigaku D-max-2550/PC diffractometer(Rigaku Inc., Tokyo, Japan). Scanning electron microscopy(SEM)(Hitachi S-3400N scanning microscope, Hitachi, Japan) was used to observe the surface morphology of the loofah and MML.

### 2.4. Adsorption of Cu^2+^

A standard stock solution of Cu(II) ions (1000 mg/L) was diluted using deionized water to obtain the desired concentration. The expected dose of MML was added to a series of 50 mL tubes containing the desired concentration of Cu^2+^ in an aqueous solution, and the adsorption process took place in a shaking water bath. In the process of adsorption, the tube containing the adsorbent and Cu^2+^ was shaken at 160 rpm in a stable temperature horizontal shaking bath. After adsorption, the adsorbent was separated from the solution using a high-speed centrifuge, and the residual concentration of Cu^2+^ was determined using aflame atomic absorption spectrometer (Z-2000, Hitachi Limited Company, Tokyo, Japan). All experiments were carried out with three replicates.

The adsorption capacity (Q_e_) of MML and loofah at adsorption equilibrium can be obtained from the following equation:(1)$${\;Q}_{e}=V\times \left(C_{0}-C_{e}\right)/M$$
where C_0_ (mg/L)and C_e_ (mg/L) are the concentration of Cu(II) ions before adsorption and at adsorption equilibrium, respectively, V (L) is the volume of the adsorbed solution, and M (g) is the mass of adsorbent added.

The experiment on the effect of pH was carried out using Cu(II) ions solution in the pH range of 1.0 to 5.0 at 25 °C. Adsorbent (0.02 g) was placed in a 50 mL tube containing Cu(II) ions solution (40 mL) at an initial concentration of 200 mg/L and an adsorption time of 120 min. The solution pH was adjusted with 0.1 M NaOH or 0.1 M HCl as appropriate.

The influence of the amount of adsorbent on the removal of Cu(II) ions was carried out under an initial Cu(II) ions concentration of 200 mg/L at pH 5.0 and 25 °C, with an adsorption time of 120 min. The range of adsorbent dosage was from 0.15 g/L to 0.75 g/L.

Isotherm studies were performed by placing 0.02 g of adsorbent into a series of tubes containing 40 mL Cu(II) ions solution with different initial concentrations in the range of 50–900 mg/L at pH 5.0 and 25 °C for 120 min.

The adsorption kinetics were investigated under an initial Cu(II) ions concentration of 200 mg/L at pH 5.0 and 25 °C. Samples of the solution(10 mL) were taken at different time intervals (5–180 min) to determine the current Cu(II) ions concentrations.

### 2.5. Adsorption and Desorption

In this part, the recyclability of MML was explored. The MML-adsorbed Cu(II) ions were added to a 0.1 M HCl solution and were then desorbed at 25 °C for 2.0 h. After desorption, the MML continues the adsorption experiment, and then desorption. Under the same experimental conditions, five consecutive adsorption–desorption experiments were carried out.

## 3. Results and Discussion

### 3.1. FTIR Analysis of MML

The FTIR spectrum of the original loofah (a), loofah-g-GMA (b), and MML (c) are shown in Figure 2. The peak near 3300 cm^−1^ is attributed to hydroxyl groups. The peaks at around 2900–2980 cm^−1^ are attributed to the C–H in methylene and methyl groups, and the peak at 1240 cm^−1^ can be assigned to the C–O in phenols, ether, or alcohols [21]. Comparing the loofah spectrum with that in Figure 2b, the peak in the range of 1726 cm^−1^ can be attributed to the carbonyl group in GMA and the peaks at 917 cm^−1^ and 821 cm^−1^ are assigned to the epoxy group in GMA [23]. The FTIR of MML is shown in Figure 2c. The characteristic peak of the epoxy groups at 917 cm^−1^ shows that the 821 cm^−1^ peak is no longer evident, which suggests that the opening ring reaction between the epoxy groups and mannitol has occurred, confirming that loofah-g-GMA has grafted withmannitol.

### 3.2. XRD Analysis of MML

XRD patterns of loofah, loofah-g-GMA, and MML are shown in Figure 3. The loofah shows scattering at 2θ = 15.0°, 22.6°, and 34.9° which is attributed to the characteristic peaks of the loofah [22]. The loofah-g-GMA shows scattering at 2θ = 24.3°, 37.2°, and 38.8° which is attributed to the characteristic peaks of PGMA. The absence of the peak at 2θ = 24.3°, 37.2°, and 38.8° in MML confirms that the crystallinity has decreased, which indicates that mannitol grafted on loofah-g-GMA destroyed the ordered structure of loofah-g-GMA.

### 3.3. SEM Image Analysis

The surface morphology of the loofah and MML were observed with SEM (Figure 4). Figure 4a shows that the surface of the loofah is smooth, while the surface of MML (Figure 4b) is shown to be uneven and the surface area increaseswith the many protrusions that facilitate the adsorption of Cu(II) ions. Figure 4c shows that the rough surface of MML is filled, which shows that Cu(II) ions are adsorbed onto the MML.

### 3.4. Adsorption Behavior at Different pH

As is known, the solution pH is the most important factor because of the effect it has on the surface charge of adsorbents, the degree of ionization, and the speciation of adsorbates [24]. Figure 5 shows that there is a great dependence between the adsorption capacity of MML and the pH of the solution [12]. The effect of initial pH on the solution in the removal of Cu(II) ions was determined over the pH range of 1.0–5.0. As the acidity of the solution increased, the adsorption capacity decreased. The adsorption capacity of MML increased from 175.73 mg/g to 556.48 mg/g when the pH was increased from 1.0 to 5.0.

The adsorption of Cu^2+^ by MML was hindered by H^+^ in the solution. As the acidity of the solution decreases, the adsorption capacity increases. If pH > 5, then OH^-^ in the solution will combine with Cu^2+^ to form Cu(OH)_2_ in precipitation, which will affect the adsorption results. Compared to the unmodified loofah, the adsorption capacity has been greatly improved and the increase was more significant when the solution was at a high pH value. The reason for this could be the introduction of more hydroxyl groups on the surface of the loofah by the mannitol modification, which increased the adsorption active point.

### 3.5. Adsorption Behavior at Different Adsorption Doses

The amount of adsorbent is also one of the factors affecting the adsorption performance. The relationship between MML adsorption capacity and the amount of adsorbent is shown in Figure 6. When the dosage of adsorbent was gradually increased from 0.15 g/L to 0.75 g/L, the adsorption capacity of MML decreased sharply from 873.42 mg/g to 413.60 mg/g. This is because the concentration of Cu(II) ions in the solution is constant, and with the increase in the amount of adsorbent, the amount of Cu(II) ions adsorbed per unit mass of the adsorbent decreases [25]. In addition, another reason may be that as the amount of adsorbent increased, hydrogen bonds formed between the hydroxyl groups of the adsorbent (the hydroxyl group of mannitol), which weakened the complexation ability of the hydroxyl group and Cu(II) ions, leading to a lower adsorption capacity.

### 3.6. Adsorption Behavior at Different Adsorption Times

Adsorption speed is also one of the properties of the adsorbent. Figure 7 shows the change in loofah and MML adsorption capacity as a function of adsorption time. It was found that the adsorption capacity gradually increased with an extension in time, indicating that the interaction between the Cu(II) ions and functional groups on the surface of the adsorbents requires a diffusion process. When the adsorption time was 120 min, the adsorption capacity of MML reached 479.28 mg/g. The adsorption capacity tends to be constant when the adsorption time is prolonged, indicating that the adsorption reached equilibrium.

### 3.7. Adsorption Behavior at Different Initial Concentrations of Cu(II) Ions 

The initial concentration has a great influence on the adsorption capacity of the adsorbent. Figure 8 shows that there is a positive ratio between the adsorption capacity of MML and the initial concentration, that is, the adsorption capacity increased when the initial concentration increased. The reason for the analysis is that the higher the concentration of Cu(II) ions, leading to a higher driving force to overcome the diffusion resistance of Cu(II) ions between the water phase and the solid phase, the easier it is to diffuse to the vicinity of the adsorption group, increasing the collision probability between the adsorption group and Cu(II) ions, making it easier to capture. The adsorption capacity was 888.89 mg/g when the initial concentration was 900 mg/L, which was much higher than that of unmodified loofah (400 mg/g). At this time, the adsorption activity point on MML was completely occupied by Cu(II) ions, and the adsorption reached saturation.

### 3.8. Isothermal Adsorption

Isothermal adsorption can explore the nature of adsorption and predict the adsorption capacity of an adsorbent. In this experiment, the adsorption mechanism of the adsorbent was explored using the Langmuir and Freundlich model. A Langmuir isotherm assumes monolayer adsorption on a uniform surface with a finite number of adsorption sites. A Freundlich isotherm model is commonly applied to describe a heterogeneous adsorption process, i.e., adsorption which takes place on a heterogeneous surface through a multilayer adsorption mechanism [12,26].
(2)$$\mathrm{Langmuir}\;\mathrm{isotherm}:\;\frac{C_{e}}{Q_{e}}=\frac{1}{K_{L}Q_{m}}+\frac{C_{e}}{Q_{m}}$$
where Q_e_ (mg/g) is the experiment adsorption capacity of MML at the adsorption equilibrium, Q_m_ (mg/g) is the theoretical maximum adsorption capacity, C_e_ is the same as Equation (1), and K_L_ is the Langmuir isotherm coefficient. The relationship of C_e_/Q_e_~C_e_ is shown in Figure 9B, which shows that C_e_/Q_e_ and C_e_ have a good linear relationship. Table 1 shows the parameter values obtained from Figure 9B.
(3)$$\mathrm{Freundlich}\;\mathrm{isotherm}:\;\log Q_{e}=\frac{1}{n}\log C_{e}+\ln K_{F}$$
where K_F_ ((mg/g) (L/mg)^1/n^) and 1/n are Freundlich model constants. Figure 9A showed the linear fitting curve of $\ln Q_{e}$~$\ln C_{e}$, and the isotherm data obtained are also listed in Table 1.

Table 1 shows the fitting results. The data shows that the correlation coefficient obtained by the Freundlich model is 0.9858 and the correlation coefficient obtained by the Langmuir model is 0.9470. The SD (standard deviation) of the Freundlich model isless than that of the Langmuir model. The Freundlich isothermal adsorption model is more suitable for the adsorption process. The adsorption mechanism of Cu(II) ions on MML is multilayer adsorption and mainly chemical adsorption, which is related to the adsorption affinity of hydroxyl on MML.

### 3.9. Adsorption Kinetics

Kinetics is the study of evaluating the effectiveness of adsorbents in terms of adsorption rates. It can also elucidate the mechanisms involved in the process. The adsorption kinetics of Cu(II) ions onto MML was explored by the pseudo-first-order and pseudo-second-order kinetic equations [27].
(4)$$\mathrm{pseudo}-\mathrm{first}-\mathrm{order}\;\mathrm{kinetic}:\;\log(\mathrm{Qe}-Q_{t})=\log\mathrm{Qe}-K_{1}t$$
where Q_t_ is the adsorption capacity of Cu(II) ions by MML at adsorption time t (min), and Q_e_ is the same as in Equations (2) and (3). K_1_ is the corresponding rate constant.
(5)$$\mathrm{pseudo}-\sec\mathrm{ond}-\mathrm{order}\;\mathrm{kinetic}:\;\frac{t}{Q_{t}}=\frac{1}{K_{2}Q_{e}^{2}}+\frac{1}{Q_{e}}t$$
where K_2_ is the corresponding rate constant.

In the pseudo-first-order kinetic reaction, the arrival of the adsorbed solute on the surface of the adsorbent is controlled by the diffusion step, while in the pseudo-second-order kinetic reaction, the arrival of the adsorbed solute on the surface of the adsorbent is controlled by the chemisorption mechanism. According to the two models, log(Q_e_-Q_t_)~t, and t/Q_t_~t are fitted linearly as shown in Figure 10, respectively. Table 2 lists the relevant parameters obtained from the two models. The data in Table 2 showsthat the linear correlation coefficient of the pseudo-second-order dynamic model (0.9827) is larger than that of the pseudo-first-order dynamic model (0.9031), and the SD(standard deviation) of the pseudo-second-order dynamic model is less than that of the pseudo-first-order dynamic model. Therefore, it can be concluded that the adsorption process of Cu(II) ions by MML is well described by the pseudo-second-order kinetic model, indicating that the adsorption process is controlled by chemisorption, through the sharing of electrons between the Cu(II) ions and hydroxyl groupson the surface of MML.

### 3.10. Adsorption Mechanism Analysis

Combining the above results, the adsorption of Cu(II) ions on MML is mainly affected by the pH of the solution and the functional groups of MML. As mentioned above, the adsorption of Cu(II) ions on MML is more favorable at a higher pH. At this point, because the hydroxyl oxygen has a lone pair of electrons, it can be shared with the metal cation and complexation occurs to achieve adsorption. However, as the pH of the solution decreases, the hydroxyl group is protonated to form -OH_2_^+^, which occupiesthe lone pair electron on the hydroxyl group, such that the hydroxyl group loses its complexation with Cu^2+^. Meanwhile, -OH_2_^+^ has the same charge as Cu^2+^, and they are mutually exclusive, which also reduces the adsorption capacity of MML [28,29,30]. In addition to complexation, the oxygen atom in the hydroxyl group of different cellulose molecular chains also has a great tendency to participate in Cu(II) ion adsorption through chelation [31]. The results showed that the adsorption process conforms to the Freundlich model, is multilayer adsorption, and mainly chemical adsorption. In addition, according to the change in the Cu^2+^ adsorption capacity of MML under different initial concentration conditions, the maximum adsorption capacity of MML can be up to 888.89 mg/g. The adsorption capacity of modified loofah for Cu^2+^ is better than that of modified natural polymers. The adsorption capacity of several modified natural polymers in the literature is listed in Table 3. The increase in MML adsorption capacity is mainly determined by the number of functional groups. A greater number of adsorption activity points(hydroxyl groups) were introduced to the surface of MML by the modification withmannitol; the molecular chain of mannitol is hydrophilic and can be fully extended in aqueous solution, which increases the chance of complexation of hydroxyl and Cu^2+^, leading to the increase in adsorption capacity. The proposed mechanism for the adsorption of Cu(II) ions onto MML is illustrated in Figure 11.

### 3.11. Adsorbent Reuse

To test the cyclic regeneration of MML, the adsorption/desorption process was explored. The efficiencies of removing Cu(II) ions by MML for five consecutive adsorption/desorption cycles are presented in Table 4. The results prove that the efficiency of removing Cu(II) ions from MML remains above 90%.

## 4. Conclusions

A low-cost, higher-efficiency adsorbent (MML) was successfully prepared by grafting mannitol onto loofah and applied for the removal of Cu(II) ions from an aqueous solution. The results showed that the Cu(II) adsorption on MML was highly pH-dependent, and MML had an excellent adsorption capacity for Cu(II) ions when the pH was 5.0, which is better than that of other reported modified materials. The adsorption isotherms showed that the adsorption of Cu(II) ions on MML resulted in a multilayer and was a dynamic chemisorption process. The adsorption behavior of Cu(II) ions conforms to quasi-second-order kinetics, suggesting chemical adsorption involving valence forces through the sharing of electrons and complexation between Cu(II) ions and MML. The results demonstrated that MML can be reused. Finally, MML obtained by modifying cheap agricultural waste belongs to resource reuse and has very good application prospects.

## Figures and Tables

**Figure 1 polymers-14-04883-f001:**
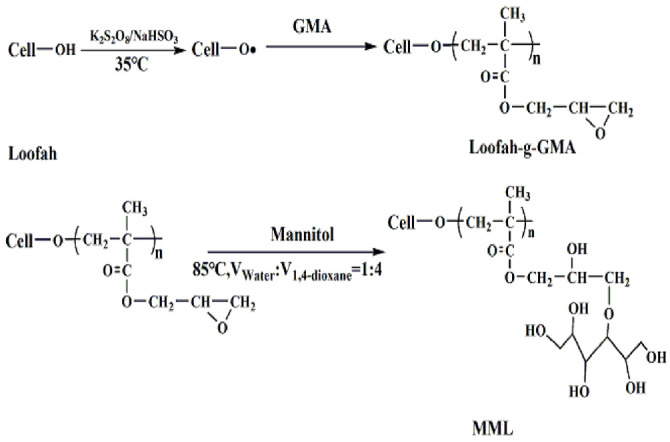
The principle of MML preparation.

**Figure 2 polymers-14-04883-f002:**
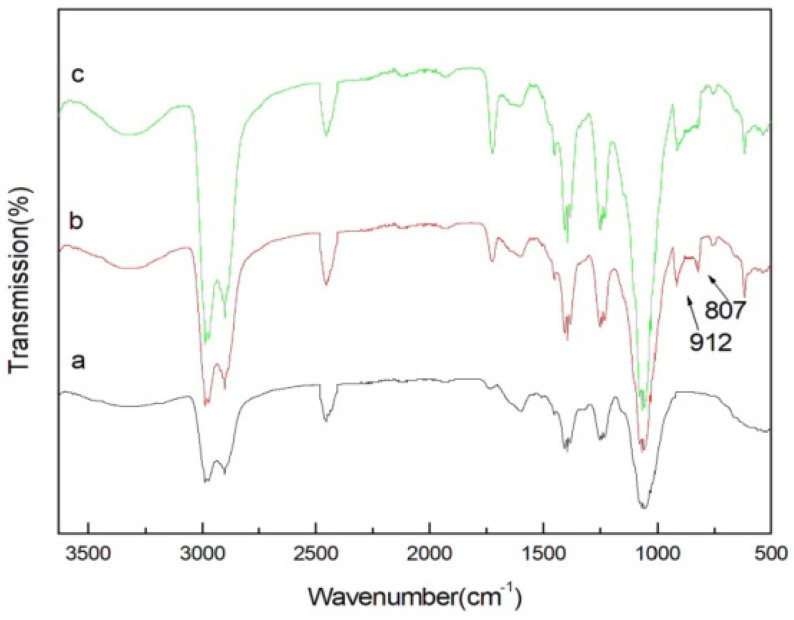
FTIR spectrum of loofah (**a**), loofah−g−GMA (**b**), and MML (**c**). (Note: the arrows point to the position of characteristic peak).

**Figure 3 polymers-14-04883-f003:**
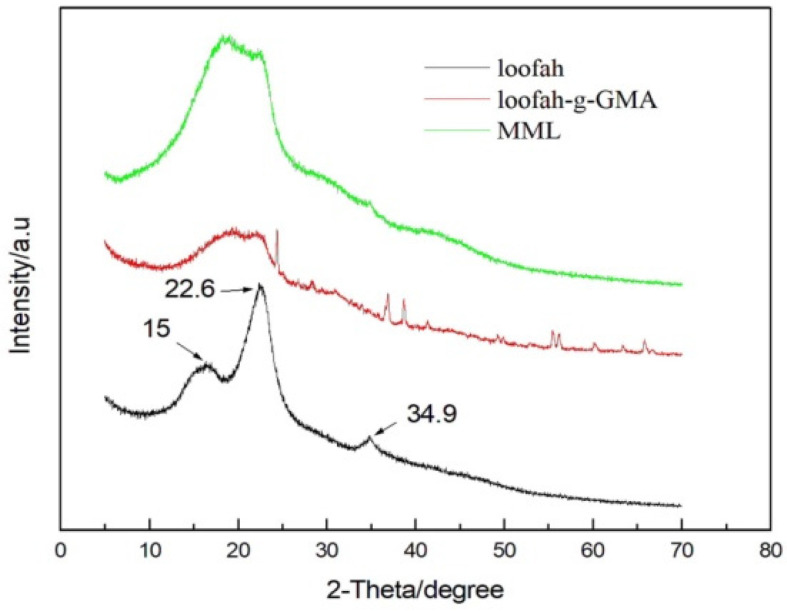
XRD patterns of original loofah, loofah-g-GMA and MML. (Note: The arrows point to the position of the diffraction peak).

**Figure 4 polymers-14-04883-f004:**
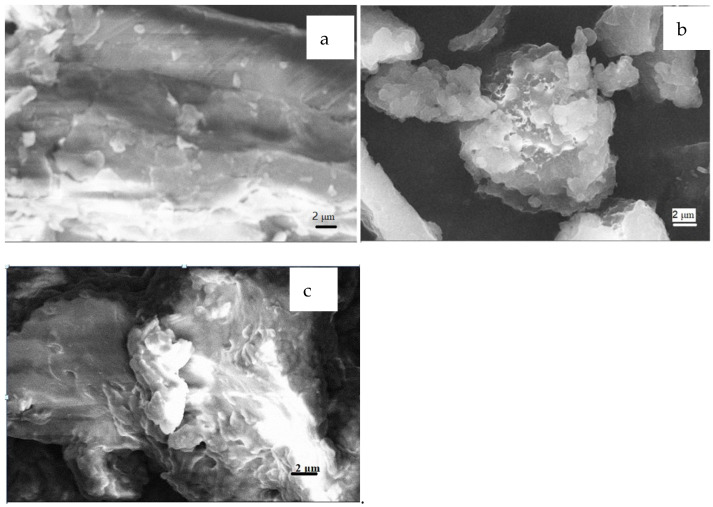
SEM images of original loofah (**a**), MML (**b**), and MML-Cu(II) (**c**).

**Figure 5 polymers-14-04883-f005:**
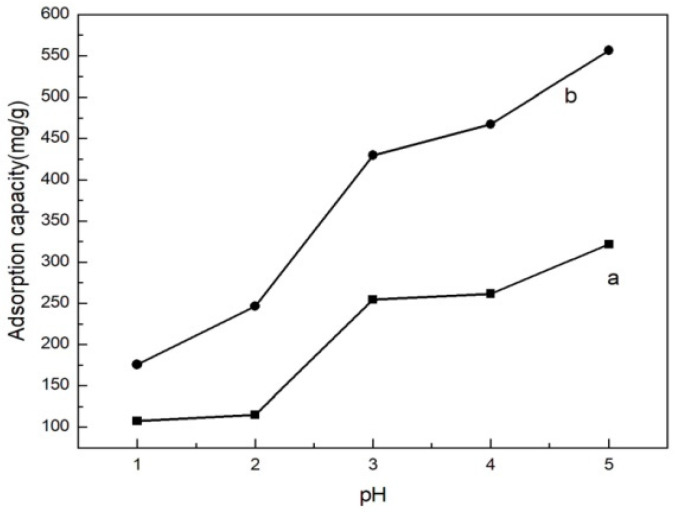
The adsorption capacity of Cu(II) ions by loofah (a), and MML (b) as a function of pH.

**Figure 6 polymers-14-04883-f006:**
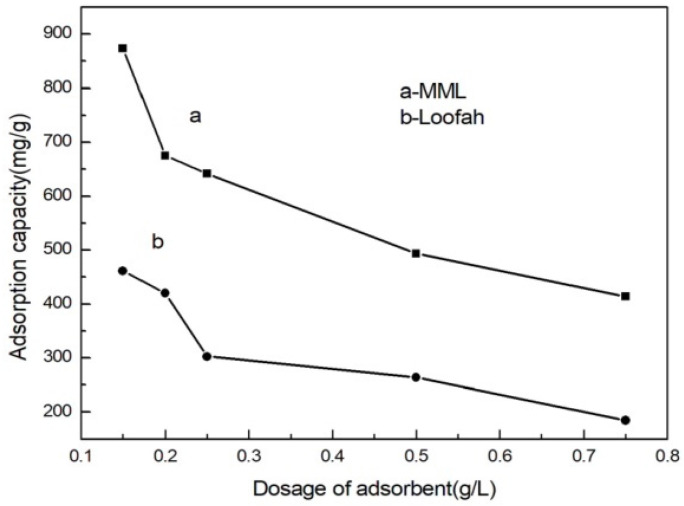
Relationship between the adsorption capacity of Cu(II) ions and the adsorption dose.

**Figure 7 polymers-14-04883-f007:**
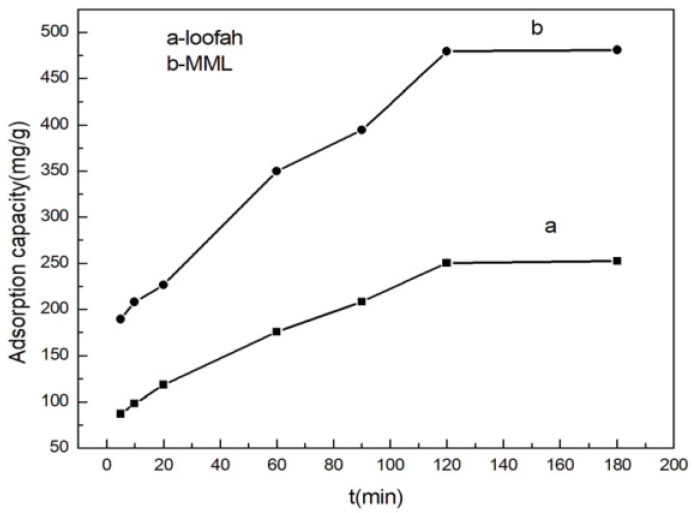
Relationship between adsorption capacity and adsorption time.

**Figure 8 polymers-14-04883-f008:**
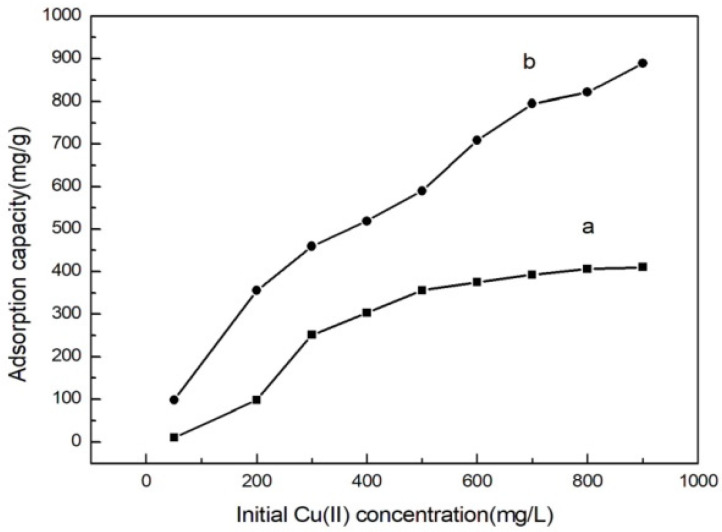
Adsorption behavior of loofah (**a**), and MML (**b**) at different initial concentrations of Cu(II) ions.

**Figure 9 polymers-14-04883-f009:**
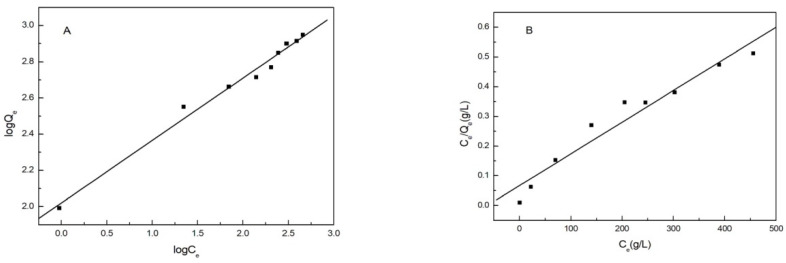
Isotherm models for Cu(II) ions adsorption onto MML: (**A**) Freundlich, (**B**) Langmuir.

**Figure 10 polymers-14-04883-f010:**
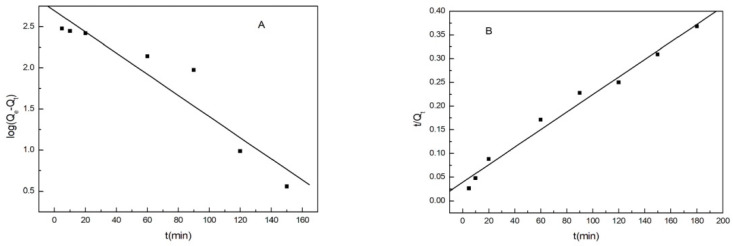
Adsorption kinetics of Cu(II) ions on MML: (**A**) pseudo-first-order kinetic, (**B**) pseudo-second-order kinetic.

**Figure 11 polymers-14-04883-f011:**
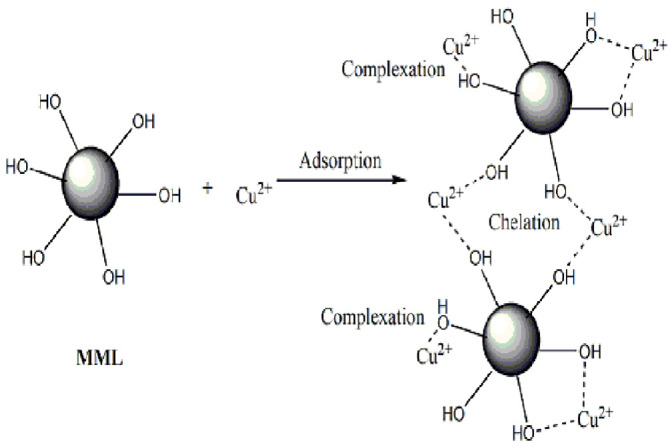
Mechanism of Cu(II) ion adsorption onto MML.

**Table 1 polymers-14-04883-t001:** Adsorption isotherm parameters of Cu(II) ions on MML.

Q_e_*/mg·g^−1^	Langmuir				Freundlich			
	Q_m_**/mg·g^−1^	K_L_/L·mg^−1^	R^2^	SD	K_F_	n	R^2^	SD
888.89	934.58	0.0161	0.9470	0.043	104.80	2.900	0.9858	0.037

* Experimental data.** Calculated data.

**Table 2 polymers-14-04883-t002:** The data of Cu(II) ions adsorption on MML obtained by two models.

Q_e_^*^/mg·g^−1^	Pseudo-First-Order Kinetic Model				Pseudo-Second-Order Kinetic Model			
	k_1_/min^−1^	Q_e_^**^/mg·g^−1^	R^2^	SD	k_2_/g·mg^−1^·min^−1^	Q_e_^**^/mg·g^−1^	R^2^	SD
489.01	0.030	495.52	0.9031	0.263	0.0001	540.54	0.9827	0.018

^*^ Experimental data. ^**^ Calculated data.

**Table 3 polymers-14-04883-t003:** Adsorption capacity of several modified natural polymers for Cu(II) ions.

Adsorbents	Adsorption Capacity (mg/g)
Amino-modified carboxymethyl chitosan [32]	175.56
Thiosemicarbazide-modified polyvinyl alcohol [33]	82.36
CS-g-AOPAM [34]	215.5
MML(in this paper)	888.89

**Table 4 polymers-14-04883-t004:** Reuse of MML.

Desorption Times	1	2	3	4	5
Removal efficiency/%	97.48	96.21	95.38	94.91	93.82

## Data Availability

The data presented in this study are available on request from the corresponding author.

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
