# Peer review of "Preparation of Mannitol-Modified Loofah and Its High-Efficient Adsorption of Cu(II) Ions in Aqueous Solution"

_polymers, 2022, doi:10.3390/polym14224883_

Round 1
Reviewer 1 Report
The manuscript describes a potential new adsorbent for copper ions removal from aqueous solutions. However, in its present form, it contains a significant deficiency and does not meet the requirements for this type of work. the list of the most important comments is provided below:
- English requires a thorough correction
- Abstract: it is not correctly constructed: the second sentence duplicates the information from the first; 'test methods" is not an inappropriate wording for a characterization technique; "The results indicated that the adsorption capacity of MML was greatly improved" - in regard to what?; "Frendich model" - there is no such model;
- introduction: it is improperly prepared, contains a negligible amount of information, is very short, and the literature review is definitely insufficient; additionally, no information explaining what loofah is, and no literature references; moreover, the purpose of the studies insufficiently described;
- 2.3 characterization methods: "Functional group information was characterized by Fourier transform infrared" - something is missing;
- 2.4 adsorption procedures should be described in detail;
- Figure 2: poor quality and resolution, no description of the axis in English;
- 3.2: there is no confirmation about the characteristic XRD pattern of loofah; the discussion is not supported by any literature reference;
- 3.3 SEM images did not confirm the tubular structure, while the Authors indicated it in the introduction; additionally, there is no discussion about the material studied particles' size;
- 3.4- 3.7: the organization of these subsections is inappropriate; there is no logical introduction here, it is not known e.g. what the pH range was, how the solutions were prepared, and there is no mention of reference material anywhere before; the same is the case for the influence of the adsorbent dose, time and concentration;
- isotherm studies: no explanation of the assumptions of both models, no unit for the Freundlich constant; correlation coefficient is not enough to assign a given model - further analysis is necessary (e.g. regression analysis for chi-square, or root mean square error);
- kinetic models: the same as in the case of isotherms;
- what about other ions? studies with other ions are necessary to determine the selectivity of the material;
- the comparison with other adsorbents must be included;
- adsorption mechanism: the discussion should be supported by physicochemical analysis of the material after adsorption;
- conclusion section contains a factual error "the adsorption process conforms to Freundlich model and is monolayer adsorption";
- the manuscript lacks confirmation of the novelty.
Author Response
- English requires a thorough correction
Answer: We'll do our best to revise the language.
- Abstract: it is not correctly constructed: the second sentence duplicates the information from the first; 'test methods" is not an inappropriate wording for a characterization technique; "The results indicated that the adsorption capacity of MML was greatly improved" - in regard to what?; "Frendich model" - there is no such model;
Answer: this has been corrected.
- introduction: it is improperly prepared, contains a negligible amount of information, is very short, and the literature review is definitely insufficient; additionally, no information explaining what loofah is, and no literature references; moreover, the purpose of the studies insufficiently described;
Answer: Thank you very much for your suggestion and this has been revised.
- 2.3 characterization methods: "Functional group information was characterized by Fourier transform infrared" - something is missing;
Answer: this has been revised.
- 2.4 adsorption procedures should be described in detail;
Answer: this has been revised.
- Figure 2: poor quality and resolution, no description of the axis in English;
Answer: this has been revised.
- 3.2: there is no confirmation about the characteristic XRD pattern of loofah; the discussion is not supported by any literature reference;
Answer: this has been revised.
- 3.3 SEM images did not confirm the tubular structure, while the Authors indicated it in the introduction;
Answer: The tubular structure described here is the macroscopic structure of the loofah.
additionally, there is no discussion about the material studied particles' size;
Answer: Thank you very much for your suggestion. Unfortunately, We have only given that powders of loofah before modified was smaller than 74m. we will try to discuss about the material studied particles' size in a follow-up study.
- 3.4- 3.7: the organization of these subsections is inappropriate; there is no logical introduction here, it is not known e.g. what the pH range was, how the solutions were prepared, and there is no mention of reference material anywhere before; the same is the case for the influence of the adsorbent dose, time and concentration;
Answer: this has been revised.
- isotherm studies: no explanation of the assumptions of both models, no unit for the Freundlich constant;
Answer: this has been revised.
correlation coefficient is not enough to assign a given model.- further analysis is necessary (e.g. regression analysis for chi-square, or root mean square error);
Answer: Thank you very much for your suggestion. The interpretation of the isothermal models is based on reference [20,25,26]
- kinetic models: the same as in the case of isotherms;
Answer: Thank you very much for your suggestion.The interpretation of the kinetic models is based on reference [20,25,26]
- what about other ions? studies with other ions are necessary to determine the selectivity of the material;
Answer:Thank you very much for your good suggestion. We will further discuss the adsorption properties of MML for other ions in a follow-up study.
- the comparison with other adsorbents must be included;
Answer: this has been revised.
- adsorption mechanism: the discussion should be supported by physicochemical analysis of the material after adsorption;
Answer:Thank you very much for your suggestion. We will consider it further in the following study.
- conclusion section contains a factual error "the adsorption process conforms to Freundlich model and is monolayer adsorption";
Answer: this has been corrected.
- the manuscript lacks confirmation of the novelty.
Answer: The innovation of this research is that, GMA was used as monomer, and a large number of epoxy groups were introduced into loofah pulp by free radical grafting polymerization, which greatly increased the probability of functionalization with mannitol, it is more advantageous to increase the amount of hydroxyl, and then improve the adsorption capacity of loofah.

Reviewer 2 Report
What is the real novelty of this paper? Experimental design used in the present studies report the kind of experiments similar to most adsorption studies published previously. There is no novelty in experimental approach.
Title: needs better formulation
English editing should be done throughout the manuscript.
Abstract: rewrite important methods and important results/findings only
Some work has been done on the removal of metal ions and likewise other adsorption. Authors should do proper study, add proper research gap and aims and objective of the paper in the introduction
A few typo errors are there. Lower case, upper case, units are not systematic and unified.
Characterization is not complete, Provide better quality figures. Should be arranged in a proper way and systematic; better to have in terms of before and after adsorption process
There are too many less-important information and discussion in the main text.
There is no any cost analysis in the manuscript
Provide comparison table with other similar materials
Provide and modify the reaction mechanism of the process in section 3.10
State main findings in the conclusions
Author Response
What is the real novelty of this paper? Experimental design used in the present studies report the kind of experiments similar to most adsorption studies published previously. There is no novelty in experimental approach.
Answer: The innovation of this research is that, GMA was used as monomer, and a large number of epoxy groups were introduced into loofah by free radical grafting polymerization, which greatly increased the probability of functionalization with mannitol, it is more advantageous to increase the amount of hydroxyl, and then improve the adsorption capacity of loofah.
Title: needs better formulation
Answer: Thank you very much for your suggestion. The title has been revised.
English editing should be done throughout the manuscript.
Answer: Thank you very much for your suggestion. We did our best to modify it.
Abstract: rewrite important methods and important results/findings only
Answer: Thank you very much for your suggestion. We did our best to modify it according to your suggestion.
Some work has been done on the removal of metal ions and likewise other adsorption. Authors should do proper study, add proper research gap and aims and objective of the paper in the introduction
Answer: Answer: Thank you very much for your suggestion. We did our best to modify it according to your suggestion.
A few typo errors are there. Lower case, upper case, units are not systematic and unified.
Characterization is not complete, Provide better quality figures. Should be arranged in a proper way and systematic; better to have in terms of before and after adsorption process
Answer: Thank you very much for your suggestion. We did our best to modify it according to your suggestion.
There are too many less-important information and discussion in the main text.
Answer: Thank you very much for your suggestion. We did our best to modify it according to your suggestion.
There is no any cost analysis in the manuscript
Answer: Thank you very much for your suggestion. Unfortunately, we will try to discuss about cost analysis in a follow-up study.
Provide comparison table with other similar materials
Answer: this has been added.
Provide and modify the reaction mechanism of the process in section 3.10
Answer: this has been added.
State main findings in the conclusions
Answer: this has been added.

Reviewer 3 Report
In this paper, the author grafted mannitol onto loofa ladle and successfully prepared a low-cost and high-efficiency adsorbent. FTIR, XRD and SEM were used to characterize the structure. After investigating the adsorption behavior of MML for cu (II) ions under different conditions, it was concluded that MML had a good adsorption capacity for cu (II) ions when the PH value was 5. The adsorption process conforms to frendich model, and the MML has good regeneration performance. The author has done a very systematic work, however, there are still some issues to be addressed. A minor revision is suggested before its acceptance.
1. At the beginning of the abstract, one or two sentences are suggested to present the background of this work.
2. More background should be added to show the nolvety of this work. In addition, some of the paragraphs can be combined together.
3. One blank space should be added between the value and unit.
4. One scheme is suggested to show the whole preparation procedure.
5. It is suggested to adjust the frame structure of chapter 3 to make it easier for readers to read.
6. The format of the picture needs to be modified and unified, and further processing is required in all figures to have a better readability and resolution.
7. It is recommended to compare with adsorption of heavy metals to highlight the advantages. The following articles could be considered for comparison: A wood-mimetic porous MXene/gelatin hydrogel for electric field/sunlight bi-enhanced uranium adsorption; Vacuum 189, 110229, 2021; Chemical Communications 56, 3935-3938, 2020;
8. The quality of the references is low and some of them are too old.
9. There are some spelling and grammar problems in the manuscript. English editing service should be performed before submission.
Author Response
- At the beginning of the abstract, one or two sentences are suggested to present the background of this work.
Answer: this has been added.
- More background should be added to show the nolvety of this work. In addition, some of the paragraphs can be combined together.
Answer: this has been added.
- One blank space should be added between the value and unit.
Answer: it has been revised in the manuscript.
- One scheme is suggested to show the whole preparation procedure.
Answer: it has been revised in the manuscript.
- It is suggested to adjust the frame structure of chapter 3 to make it easier for readers to read.
Answer: Thank you very much for your suggestion. It has been revised in the manuscript.
- The format of the picture needs to be modified and unified, and further processing is required in all figures to have a better readability and resolution.
Answer: it has been revised in the manuscript.
- It is recommended to compare with adsorption of heavy metals to highlight the advantages. The following articles could be considered for comparison: A wood-mimetic porous MXene/gelatin hydrogel for electric field/sunlight bi-enhanced uranium adsorption; Vacuum 189, 110229, 2021; Chemical Communications 56, 3935-3938, 2020;
Answer: Thank you very much for your suggestion. It has been revised in the manuscript.
- The quality of the references is low and some of them are too old.
Answer: it has been revised in the manuscript.
- There are some spelling and grammar problems in the manuscript. English editing service should be performed before submission.
Answer: We'll do our best to revise it.

Round 2
Reviewer 1 Report
The authors made some changes, but unfortunately, they did not follow some of my comments:
- the introduction still needs to be improved - the description of the agricultural waste application is very limited;
- I insisted on further analysis of mathematical models - isotherms and kinetics (e.g. chi-square or root mean square error);
- the comparison with other ions - it is crucial for this kind of paper;
- the discussion about the adsorption mechanism is not supported by physicochemical analysis of the material after adsorption;
Additionally, Figure 7 presents the results for MML, while it should also contain the results for the reference material.
Author Response
the introduction still needs to be improved - the description of the agricultural waste application is very limited;
Answer: Thank you very much for your suggestion and this has been revised.
- I insisted on further analysis of mathematical models - isotherms and kinetics (e.g. chi-square or root mean square error);
Answer: Thank you very much for your suggestion and this has been revised.
- the comparison with other ions - it is crucial for this kind of paper;
Answer: Thank you very much for your suggestion. In this manuscript, we only discuss the adsorption of copper ions by MML, which is the result of one stage of our research. But we believe this study can provide reference for the adsorption of other heavy metal ions. In the following study, the adsorption of various heavy metal ions and mixed ions by MML will be discussed in detail, so as to further explore the adsorption selectivity of MML for various heavy metals ions.
- the discussion about the adsorption mechanism is not supported by physicochemical analysis of the material after adsorption;
Answer: Thank you very much for your good advice. The analysis of the adsorption mechanism in this paper is inferred by us on the basis of some references [28,29,30], and it is our negligence that there is no physicochemical analysis support for the materials after adsorption. Next, we will systematically study the adsorption of other heavy metal ions by MML. The adsorption mechanism was confirmed by physicochemical analysis of MML after adsorption by FTIR, XPS and other means.
Reference[28]: Sutirman, Z. A.; AmiraRahim, E.; MarsinSanagi, M.; Karim, K. J. A.; Ibrahim, W. A. New efficient chitosan derivative for Cu(II) ions removal: Characterization and adsorption performance, International Journal of Biological Macromolecules 2020,153, 513-522
Reference[29]: Wang, J. J.; Cao, M. S.; Jiang, C. Y.; Zheng, Y. X.; Zhang, C. S.; Wei, J. Adsorption and coadsorption mechanisms of Hg2+ and methyl orange by branched polyethyleneimine modified magnetic straw, Materials Letters 2018, 229, 160–163
Reference[30]:Ranasinghe, S. H.; Navaratne, A. N.; Priyantha, N. Enhancement of adsorption characteristics of Cr(III) and Ni(II) by surface modification of jackfruit peel biosorbent, Journal of Environmental Chemical Engineering 2018, 6(5), 5670-5682
Additionally, Figure 7 presents the results for MML, while it should also contain the results for the reference material.
Answer: Thank you very much for your suggestion and this has been added.

Reviewer 2 Report
Even after the improvements made by the authors, some of the comments/answers still have doubts on the suitability of the manuscript to be published in the journal. i.e,: Characterization studies, cost analysis, comparison with similar studies. Thus, my recommendation is to reject the paper.
Author Response
Even after the improvements made by the authors, some of the comments/answers still have doubts on the suitability of the manuscript to be published in the journal. i.e,: Characterization studies, cost analysis, comparison with similar studies. Thus, my recommendation is to reject the paper.
Answer:
The adsorption properties of mannitol modified loofah(MML) were studied systematically, and some valuable conclusions were obtained. We believe that this study has important application value in the treatment of heavy metal ion wastewater.
As for the cost analysis, this study is a stage research result, and the focus is on the discussion of adsorption properties of MML. However, cost analysis is essential when evaluating the value of MML applications in the future. This is also what we need to discuss in the follow-up study.
Compared with other similar studies, our advantage or difference is that more epoxides can be introduced through free radical copolymerization, and more mannitol reacts with epoxy groups. Thus, more hydroxy groups are introduced to the surface of the loofah. This method greatly improved the adsorption capacity of loofah.

Reviewer 3 Report
Accept in present form
Author Response
Thank you very much for your recognition of this paper.

Round 3
Reviewer 1 Report
The Authors responded properly to all my comments.
Author Response

(The authors gave the same response as above.)

Reviewer 2 Report
Authors provided answers to the reviewer’s questions and revised the manuscript following the reviewer’s comments. However, I am not still sure. There is no novelty in experimental approach or any contribution to further mechanistic understanding, not recommended due to lack of real novelty, progress in scientific understanding and clarity in experimental conditions reported in materials and methods and results and discussion
Author Response
Answer:
Thank you very much for your advice. Following your suggestions, we made further revisions in the introduction, references, experiment part, the results, conclusions, and so on. We hope to get your approval for our modification, thank you very much.
The mannitol modified loofah was prepared by a new synthetic route, and the adsorption capacity of loofah was obviously improved. The loofah has a good application prospect in the field of heavy metal wastewater treatment, and the resources of loofah can be reused, increased its economic value.
Compared with other similar studies, the novelty of this study lies in the design of the synthetic route. The PGMA chain with a large amount of epoxy group was introduced on the surface of loofah by radical copolymerization with GMA monomer, more epoxides can be introduced, and more mannitol reacts with epoxy groups. Thus, more hydroxy groups are introduced to the surface of the loofah. This method greatly improved the adsorption capacity of loofah.
Additionally, mannitol is a cheap and non-toxic modifier, so the cost of preparing the adsorbent is very low.
